# Graph Theory Measures and Their Application to Neurosurgical Eloquence

**DOI:** 10.3390/cancers15020556

**Published:** 2023-01-16

**Authors:** Onur Tanglay, Nicholas B. Dadario, Elizabeth H. N. Chong, Si Jie Tang, Isabella M. Young, Michael E. Sughrue

**Affiliations:** 1UNSW School of Clinical Medicine, Faulty of Medicine and Health, University of New South Wales, Sydney, NSW 2052, Australia; 2Omniscient Neurotechnology, Level 10/580 George Street, Sydney, NSW 2000, Australia; 3Robert Wood Johnson Medical School, Rutgers University, 125 Paterson St, New Brunswick, NJ 08901, USA; 4Yong Loo Lin School of Medicine, National University of Singapore, 10 Medical Dr, Singapore 117597, Singapore; 5School of Medicine, University of California Davis, Sacramento, CA 95817, USA

**Keywords:** centrality, dMRI, eloquence, graph theory, hubness, neurosurgery, connectome, brain mapping

## Abstract

**Simple Summary:**

Advances in our understanding of human brain structure and function have been facilitated through improved mapping of the structural and functional neural connections throughout the human brain ‘connectome’. By utilizing different statistical techniques and non-invasive imaging modalities to capture the structural and functional properties of the brain connectome, such as with diffusion or functional MRI, the brain can also be represented as a graph of individual nodes which are connected throughout a network. Previously, the neurosurgical community has often relied on traditional maps of the human brain to identify highly functional regions, often called ‘eloquent’, but these regions differ between patients and do not always provide an adequate guide to reliably prevent functional deficits. Through graphically representing the brain, mathematical graph theory approaches may be able to provide additional information on important inter-individual network properties and functionally eloquent brain regions. This review attempts to outline and review the applicability of graph theory for neurosurgery.

**Abstract:**

Improving patient safety and preserving eloquent brain are crucial in neurosurgery. Since there is significant clinical variability in post-operative lesions suffered by patients who undergo surgery in the same areas deemed compensable, there is an unknown degree of inter-individual variability in brain ‘eloquence’. Advances in connectomic mapping efforts through diffusion tractography allow for utilization of non-invasive imaging and statistical modeling to graphically represent the brain. Extending the definition of brain eloquence to graph theory measures of hubness and centrality may help to improve our understanding of individual variability in brain eloquence and lesion responses. While functional deficits cannot be immediately determined intra-operatively, there has been potential shown by emerging technologies in mapping of hub nodes as an add-on to existing surgical navigation modalities to improve individual surgical outcomes. This review aims to outline and review current research surrounding novel graph theoretical concepts of hubness, centrality, and eloquence and specifically its relevance to brain mapping for pre-operative planning and intra-operative navigation in neurosurgery.

## 1. Introduction

Preserving brain function and maximizing quality of life are fundamental goals in neurosurgery while increasing the extent of resection. Advances in surgical methods, including intraoperative mapping, awake surgery, and neurophysiology have helped minimize post-surgical neurological deficits and improve patient outcomes [1,2]. Generally, these techniques attempt to identify brain areas which are more readily associated with observable functions to inform the surgeon of which areas can be cut safely. Collectively referred to as “eloquent” brain, previous attempts to delineate these regions preoperatively through the use of imaging techniques such as functional magnetic resonance imaging (fMRI) have not been widely implemented, potentially due to unavailability of pipelines for clinical translation [3,4]. 

The application of graph theory to the brain is a promising research area which has potential to further improve patient safety [5,6]. Combining non-invasive imaging with statistical modeling, the brain can be represented as a network of elements, or nodes, with pairwise connections, or edges [7]. The complete set of these nodes and edges comprises the adjacency matrix of the connectome, a model of the brain topology [8]. Several metrics defined within the framework of graph theory can then be applied to this graph to examine the information flow within the brain, in health and disease (Table 1). There is still a limited understanding of the neurological correlates and clinical utility of these mathematical measures [9]. 

## 2. Connectomics of the Brain

Graph theory provides a mathematical representation of brain architecture, consisting of nodes and edges. The definition of nodes and edges depends on the scale and technique used to study the brain. Broadly, graphs may model structural or functional connectivity based on a group of brain regions, known as a brain network. Graph theory has been popular in connectomics, which is defined as the study of the anatomical and functional connections between regions in the brain.

### 2.1. Graphing Structural Connectivity

Structural connectivity at the macroscale relies on diffusion MRI (dMRI), which is based on the water molecule diffusion in the brain. When unconstrained, water molecules have an equal probability of diffusing in any direction, referred to as isotropic diffusion [10]. In the brain however, axons constrain water molecule diffusion, resulting in anisotropic diffusion parallel to fibre direction [11]. The tracts are reconstructed using fibre tractography, which propagates streamlines based on the direction of water diffusion [12]. The magnitude of diffusion can be measured using dMRI. A model is then applied, most commonly diffusion tensor imaging (DTI), to infer the direction of diffusion within each voxel [13]. DTI however performs poorly in estimating multiple orientations, and therefore crossing fibres, which account for 30–90% of white matter volume in the brain [14]. Other methods, such as constrained spherical deconvolution, or Q-ball imaging are increasing in popularity to address these concerns. 

Once streamlines have been tracked between all nodes, the strength of connectivity, or the edge weight is derived. As deterministic tractography is the most commonly utilized method, the number of streamlines, or connections, between each node is the most common measure of edge weight. Streamlines, however, do not directly correspond to axons and are confounded by factors unrelated to connectivity such as myelination, density, and axonal number within a region [15,16]. 

The inherent limitation, and a popular criticism against dMRI-based tractography is the lack of ground truth. Attaining actual ground truth for the human brain however would require a tremendous effort to track every single axon, which is currently unlikely to be achieved. Despite concerns over sensitivity, the utility of tractography has been validated in several neurosurgical studies [17,18,19], and intraoperative visualization tools are becoming increasingly popular. Any improvements in the accuracy of these techniques will further benefit operative planning. 

Finally, dMRI is unable to provide the direction of information flow, and therefore graphs based on diffusion tractography are undirected (Figure 1), resulting in incomplete representation of network structure [20]. Nonetheless, dMRI remains the best available method to study structural connectivity, and methods to improve its accuracy are an area of active research [21,22,23]. 

### 2.2. Graphing Functional Connectivity

In contrast, functional connectivity reflects regions which co-activate at a given time. While the nodes may be similarly defined as structural graphs, edges in functional graphs are often a correlation coefficient, summating neurophysiological activity over time. Depending on the desired temporal resolution, functional connectivity can be measured non-invasively using fMRI, electroencephalography (EEG), or magnetoencephalography (MEG). EEG uses electrodes placed on the scalp to measure electrical activity which is thought to arise from synchronized activation of pyramidal neurons [24]. MEG on the other hand uses superconducting quantum interference devices (SQUIDs) to detect magnetic fields generated by neuronal currents [24]. While EEG and MEG display millisecond changes in functional connectivity, they lack spatial precision since their sensors receive aggregate activity over large populations of neurons [25,26]. As a result, it becomes difficult to precisely localize connectivity changes. This is compounded by volume conduction, which refers to the effect of measuring electrical activity away from its source, as the signal is conducted through several tissue compartments of the head [24]. Conversely, fMRI offers lower temporal resolution at around 0.5 to 1.5 Hz [27]; however, due to its higher spatial resolution, it is the most common method of acquisition for functional connectivity studies. 

fMRI relies on fluctuations of the blood-oxygenation-level-dependent (BOLD) signal based on the differences of magnetic properties between oxygenated and deoxygenated blood [28]. fMRI is therefore a haemodynamic measure of neuronal metabolic demand in a given region, and edges of the functional graph produced from fMRI are Pearson correlation coefficients of the BOLD signal between two regions. Fluctuations in the BOLD signal have been shown to correlate with neuronal dynamics [29]; can influence task-evoked activity and behavior [30,31]; and are under genetic control [24,32,33]. Resting-state fMRI has also been shown to correlate with structural connectivity measured with diffusion MRI and tract tracing studies [24]. Conversely, since it is an indirect measure, it is unclear whether an increase in the BOLD signal is due to excitation or inhibition in an area, though the correlation coefficient between the BOLD signal in different areas has been shown to be a sensitive method of detecting network connection in experiments using simulated BOLD signals on several ground-truth networks [34]. Moreover, given its hemodynamic nature, the BOLD signal can be confounded by variations in vasculature [35] and ageing [36]. Studies are however underway to improve the measurement of functional connectivity, either through post-processing methods to improve reproducibility, or through multi-modal measurement techniques combining MEG or EEG with fMRI, in order to advantage from the temporal and spatial resolution of each [37,38]. Despite its disadvantages, fMRI remains the most widely used measure of functional connectivity as it is able to provide insight into the whole-brain connectome and is currently more widely available than MEG [39]. 

## 3. Parcellating the Brain

The measurement of network connectivity is affected by the way regions of the brain are delineated to define nodes, which are also known as parcellation [40,41]. Traditionally, parcellations have been based on anatomical landmarks and cytoarchitecture, with the latter relying most basically on the distribution of neuronal cell bodies, which is fundamental to Broddmann’s parcellation system [42,43]. These methods commonly rely on microarchitecture, making it difficult to apply without autopsy [43]. Furthermore, while these maps enable a simple schema to understand neurobiology and pathology, they fail to incorporate the functional organization of the brain, which may lead to poorly defined nodes. For example, the anterior cingulate cortex is often represented as a single region in many atlases. This region however demonstrates enough heterogeneity in structural and functional connectivity to warrant further dividing it into distinct areas [44]. Since many of these atlases are based on small samples, they also neglect individual differences which may exist [45]. 

Recent approaches to parcellation incorporate functional connectivity to separate regions more accurately. The Glasser atlas is a cortical parcellation scheme generated from imaging data of 210 adults [46]. It employs a machine-learning classifier to parcellate regions based on resting state fMRI for functional connectivity, task functional MRI for cortical function, myelin content, and cortical thickness. A recent systematic review comparing different parcellation measures concluded that an optimal method to parcellate the cortex could not be determined since they each utilized different methods and subjects [47]. However, it is recommended that parcellation schemes based on multiple modalities, such as the Glasser scheme, should be prioritized as they are more accurate when analyzing functional connectivity. Another study, however, demonstrated that parcellations based on functional connectivity may change based on the task performed during fMRI [48], though the implications of this have not been studied. Therefore, while further studies are required to produce optimal cortical parcellations for structural and functional graphs, the Glasser scheme is a validated approach which consistently outperforms other methods. Unfortunately, the machine learning tool used for the Glasser atlas is not publicly available and only group averages from the Glasser study may be utilized. It must be noted that these group averages may misleadingly dismiss important individual anatomical differences and therefore further study of patient or subject-specific machine learning tools is essential [6,49]. Machine learning tools may enable patient-specific automated parcellation based on multi-modal features, thereby accounting for differences in brain architecture, and even enabling parcellation of anatomically distorted brains, which may for example be useful when studying patients with brain tumors [50]

## 4. Hubness and Centrality

Once a network is constructed, measurements can be made on the topology of the network (Figure 2). Within a network, nodes which have structural or functional significance are referred to as network hubs [51]. There are a number of mathematical approaches to define hubs within a structural or functional network [52]. However, centrality—or the ability of a node to influence or be influenced by other nodes as a result of its connection topology—is the most common measurement [53]. There are several measures of centrality which have been used to analyze the brain (Figure 3). 

### 4.1. Types of Centrality

Degree centrality is the simplest measure of centrality. The degree of a node corresponds to the number of edges connected to that node [55]. However, this treats all connections equally, without considering the influence of each connection. In contrast, eigenvector centrality accounts for the quality and quantity of connections by including the degree of the neighbors of a node in the calculation [6,56]. A similar measure, PageRank centrality, scales the influence of a node’s neighbors by their degree [57]. This minimizes bias when calculating centrality for nodes which may be connected to a single high-degree node. PageRank centrality has been considered as a measurement to identify important hubs for the planning of supratentorial neurosurgery [57,58]. 

In contrast to degree-based measures of centrality, betweenness and closeness centrality rely on shortest paths between nodes. Betweenness centrality identifies the fraction of shortest paths between any pair of nodes which pass through the selected node, or the extent to which it lies between all pairs of nodes [59]. It quantifies how many times a node acts as a bridge of information transfer to other regions. In contrast, closeness centrality is a measure of the average distance between the selected node and all other nodes [53]. 

The aforementioned centrality measures may not be suitable for functional graphs. Path length in a functional network is difficult to define, while node degree is biased by the size of the communities they belong to [60]. Communities in a graph refer to groups of nodes which are highly connected to each other compared to other nodes in the graph [61]. Once the graph has been partitioned into communities, a proposed method to characterize hubs in functional networks is participation coefficient rather than degree-based hubs from centrality analysis [60]. Participation coefficient is a measure of the distribution of a node’s edges among the communities within the graph [62]. A participation coefficient of 0 refers to a node which only communicates with nodes in its own module, while the participation coefficient approaches 1 if the node’s edges are evenly distributed among every module in the graph. As an example, participation coefficients have been studied to be used as preoperative prediction value for cognitive decline in patients with resective neurosurgery [63]. 

### 4.2. Centrality as a Measure of Hubness

Hubness is a measure of how influential and significant a node is in relation to surrounding nodes. Different centrality measures are often correlated [53,64,65], and it is therefore possible to define network hubs by combining rankings of centrality measures, though disparities arise depending on measures chosen. Sporns and colleagues concluded in tract-tracing studies on cat and macaque brains that hub nodes had high degree, closeness, and betweenness centrality [52]. This definition of hubness was later applied to human structural connectivity graphs to conclude that the right caudate, left and right superior frontal gyrus, right middle cingulate gyrus, right precuneus, left and right putamen, and left thalamus were the highest-ranking hubs [66]. This study, however, relied only on 40 participants, which is unlikely to reflect the degree of heterogeneity within the general population. In fact, a recent study demonstrated unexpected hubness at the individual level using PageRank centrality in areas which were lowly-ranked at the group level [58]. Evidently, there is no validated method to characterize hubness, and group-based averaging cannot account for interindividual variation. Nonetheless, hub regions are often association regions which do not have direct functional association, making it difficult to map during surgery. 

Modeling hubness based on distance-based centrality measures rests on the assumption that communication in the brain occurs through shortest paths. This presupposes that neurons have access to information about global topology when sending information [67]. Since this may be unlikely, other models of neural communication have recently emerged. Among these, diffusion proposes that neurons send information to all their neighbors in parallel [68]. Centrality measures based on a diffusion model have also been proposed [67,69]. These measures only partially correlate with degree, closeness, and betweenness, suggesting they may offer further information about centrality [59]. Diffusion-based models however have not been adopted as widely within preclinical and clinical connectivity studies. This may be due to the relative simplicity of degree and path-length based measures, and their availability within network analysis packages. The traditional measures also remain applicable in clinical studies, and it is difficult to challenge their validity when they consistently explain empirical evidence. Further studies are necessary to determine information flow dynamics within the brain, though this may be beyond current technological capabilities. Some clues may be derived from developmental studies examining hub node formation, though results of these have so far been conflicting [70,71]. Future studies into neurosurgical eloquence may also benefit from the application of other topological indices which have been extensively used in the application of graph theory in other fields, though as is the case with any metric, deriving clinical meanings from these values remains the challenge.

## 5. Eloquence in Neurosurgery

Eloquent brain regions refers to brain regions with direct functional association, which when injured leads to neurological deficit [72]. While such classification has surely benefited the neurosurgical community previously, it is important to consider that regions outside traditional language and motor regions also maintain functional relevance [2]. Furthermore, damage to these traditionally defined eloquent regions however do not always lead to impairment [73,74,75], where in some patients, focal lesions in areas are not defined as eloquent can lead to unexpected cognitive deficits not associated with the regional function. For example, multiple-domain cognitive decline was observed in both groups in a study which attempted to compare the cognitive effects of resection of insular glioma compared to glioma in other locations [76]. Studies into the neurocognitive effects of resection are conflicting and the incidence of deficits following resection are unknown. While some studies demonstrate improvement from baseline [77,78], others report worse neurocognitive outcomes [79], and many studies suffer from methodological limitations such as inappropriate assessment tool choice and attrition bias with worse-off patients often lost to follow-up. Although recent machine learning models can accurately predict average outcomes following neurosurgery [80], the heterogeneity of intracranial pathologies and the observed individual responses to surgery require patient-level prediction models to aid in prognostication and surgical planning.

### 5.1. Individual Variability 

Disability caused by focal lesions to eloquent areas cannot be discounted, as it can certainly be predicted for instance that damage to the primary visual cortex will lead to visual impairment. Nevertheless, post-operative impairments suffered by patients are not equal, even when the surgeries are in the same areas of non-eloquence [58]. This suggests that there is inter-individual variability in cerebral eloquence, where some patients may be more eloquent than others in areas previously recognized as non-eloquent [81]. While previous areas recognized as high-eloquence areas were confirmed to have high PageRank centrality, there was also a level of inter-individual anatomical variability found [81], with unexpected hubs found in up to 8% of people in areas previously thought to be insignificant. This reinforces the need for novel approaches to identifying brain areas with heavy neuropsychological burden, where routine diffusion tractography can identify obvious functional associations, but is less able to elucidate areas of cognitive and neuropsychological eloquence. Such inter-individual variability in cerebral eloquence and therefore responses to focal lesions in neurosurgery may be addressed by expanding our traditional idea of “eloquent” regions to one which also considers the hub regions of network (Figure 4). 

### 5.2. Hubness as a Measure of Eloquence

A number of recent works suggest hub regions are most susceptible to various neurological diseases, and given their importance within a network, damage to these regions confers greater damage on the network [82,83,84]. Reasons for this are related to the observations that hubs: (1) make several long-distance connections [85,86], which are susceptible to white matter injury [87,88]; (2) they lie on many shortest paths [89], allowing pathology to spread easily to these nodes [90,91]; and (3) may have higher metabolic requirements [92], making them susceptible to metabolic stress. Empirically, cognitive recovery of ischaemic stroke patients was predicted using a score of the extent to which hub nodes were affected [93]. Infarcts in regions with higher hubness scores was associated with reduced global efficiency, while strokes in regions with lower scores were independent predictors of better cognitive function at one-year post-stroke. The odds ratios for their univariate and multivariate models of cognitive recovery suffered from wide confidence intervals approaching one. It is likely that a 75-patient sample was not large enough to observe a robust relationship. The study also suffered from selection bias, as only patients with post-stroke cognitive impairment were recruited. It therefore overlooked whether those without cognitive impairment suffered from strokes to hub nodes. Nonetheless, several studies have established a relationship between hub nodes and neurocognitive performance [94,95,96], though it is uncertain which measures are most useful to define hub nodes in a clinical setting. To compare centrality measures empirically, Warren et al. showed that 19 patients with focal lesions in high participation coefficient areas had more severe cognitive impairment compared to 11 patients with lesions in high degree centrality areas [97]. Aside from small sample size, the study lacked sufficient description of the neuropsychological assessment tools used, making it difficult to evaluate its validity. Further clinical studies are required to compare centrality measures. 

## 6. Emerging Difficulties and New Prospects Moving Forward

### 6.1. Difficulties of Analysis and Interpretation for Patients

While the applications of graph theory are not relatively new, their ability to represent brain connectivity data for clinical translation in particular has brought about new challenges which must be considered. Firstly, there is a still need for a formalized parcellation strategy to conventionally define nodes in networks [7]. The different thresholding methods as well as the reproducibility and small sample sizes of current studies limit the ability to standardize such a protocol [98]. At the current time, it likely remains most wise to use a parcellation atlas which utilizes multi-modal integration of varying data sources, such as both in- and ex-vivo data and quantitative analyses of imaging signals, in order to produce a more biologically grounded approach [46,98,99]. 

After parcellation, there remains concerns of graph theory analyses on patient connectomes because data may be produced which has been affected by a number of problems such as multiple-comparison errors due to the high-dimensionality of the data, subjective utilization of varying thresholds between pipelines to differentiate true connections from noise, and also spatial embedding concerns which introduce distance dependent errors into connectivity maps due to motion artifacts and poor preprocessing techniques [98]. A number of solutions have been proposed for these problems, such as ML based analyses to manage the high dimensionality of individualized patient data [100] and network based statistics to control for family-wise error [101]. Although it remains necessary to utilize these graph theory methods with patient data in a cautious manner which prospectively considers many of these known problems and the strengths and limitations of these analyses based on their specific goals, or else the data will not be an appropriate neurobiological representation. 

Similarly, appropriate interpretation of graph theoretical metrics has become a key challenge because many of these metrics rely on basic assumptions which may be more applicable to complex systems beyond the brain. For instance, pagerank centrality is designed for directed networks and therefore when used on a brain connectivity data the brain graph must be treated as a directed network with bidirectional edges, although this information is difficulty to estimate reliably with modern neuroimaging modalities. Even more broadly, it still remains unclear the appropriate hub measures which best estimate desired clinical correlates, such as determining if pagerank in fact best correlates with “eloquent” regions [6,57,58]. However, it must be noted this problem is largely related to the previous lack of studies which sought to link computational or experimental work with clinical outcomes, and moving forward such comprehensive work will hopefully clarify many of these questions.

### 6.2. Clinical Graph Theory on the Horizon 

Despite these concerns, the application of graph theory in a clinical setting is promising with increasing work recently demonstrated. Graph theory and its identified metrics have been identified as potential biomarkers in Parkinson’s disease [102], in the detection of an incoming seizure in the field of epilepsy [103], and in the identification of atypical hemispheric dominance and language reorganization [104]. As such, graph theoretical measures in neurological patients can provide a novel non-invasive tool which can aid in early diagnosis, surveillance of disease progression, and potential targets for therapeutic interventions [105].

A particularly interesting application with relevance to intra-axial brain surgery can be seen with percolation theory. Once hub nodes have been defined preoperatively, the effects of lesions on the network can be predicted using percolation models by deleting nodes from the graph and measuring the outcomes. In one study, removal of central hub nodes—especially those in the temporoparietal junction and the superior frontal gyrus—reduced functional connectivity across the brain in graphs derived from five adult males [106]. Although the small sample size and imprecise parcellation method limits external validity, it provides a utility for graph theory in predicting lesion effects. Since then, percolation has been studied in some pathological processes [83,107], however its clinical applicability remains unclear as they have not been validated against empirical findings. Recently, in a proof-of-concept study, Aerts et al. applied graph theory metrics to compare the functional connectivity maps of post-operative glioma patients to lesion maps produced from pre-operative scans [108]. They first performed virtual neurosurgery on structural connectivity graphs, then utilized a model to derive the functional connectivity of the lesion maps. However, their model only corresponded with the empirical results in four out of seven glioma patients, limited by small sample size and choice of anatomical parcellation. 

Although early, graph theory models provide the ability to pre-operatively predict or intra-operatively estimate a patient’s response to operative decisions, and thus can provide information to assist with surgical decision making regarding angle of approach or extent of resection. For instance, the success of cavernous malformation surgery for lesions deep to the surface is largely predicated based on the appropriate angle of surgical approach. Furthermore, while small differences in the angle of approach may seem non-trivial, inter-individual differences in patient structural–functional connections and hub regions can lead to varying unpredicted outcomes [109]. Pre-operative graph theory analyses may predict patient-specific epicenters of damage or hubs which confer the most damage when removed from a graph, and therefore can guide simple but meaningful decisions between patients. For instance, differentiating between an anterior medial vs. posterior lateral approach in the inferior parietal lobe (IPL) to the same deep seated lesion in patients with an anterior medially located hub region vs. posterior laterally located hub in the IPL. Once an approach is decided based on these connectomic maps, intraoperative neuronavigation may provide an additional tool which can assist with resection decisions by examining the functional integrity of various connectomic structures, but further work should clarify the additional clinical benefits of combining these modalities [5,110].

The inability of network studies to account for the brain’s physiological response to injury may contribute to the disparity in empirical studies. Structural plasticity of the nervous system accounts for significant long-term improvement and even complete recovery in many disorders. Although compensation can be inferred from a network model, modeling contralateral homotopic hyperexcitability, and axonal and dendritic sprouting which constitute plasticity may be beyond the capability of current models [111,112]. In a recent comparison of the functional connectivity graphs of chronic stroke participants and matching percolated graphs derived from healthy controls, percolation was able to model short-term consequences of lesions; however, the connectomes in chronic stroke had re-organized with the emergence of new network hubs [113]. Therefore, future connectivity models must account for the dynamic nature of the brain, which requires the longitudinal characterization of these changes across different pathologies. 

Quicktome, a surgical software developed to identify anatomical locations of significant hubs based on individual connectomic maps, may provide an advantage in terms of catering to inter-individual variability in cerebral eloquence. It was able to visualize pathways and assist in navigation in patients that were not suitable for awake craniotomies, providing similar guidance around important language areas [17]. This software was also found to be able to explain the previously not understood phenomenon of cognitive dysfunction variability experienced in post-operative glioma patients, in visualizing hubs that were disrupted by the glioma surgery. While these initial results show significant potential for this software as a tool to fill in the gaps in current fMRI and DTI imaging, large amounts of further development are required to allow for external generalization past its current assumptions of left-hemispheric dominance and western language utilization [17]. 

## 7. Conclusions

Currently, intraoperative monitoring remains the gold-standard in practice. However, regions which require the most attention to preserve—the hub nodes—have unclear functional association, making it difficult to immediately observe deficits intraoperatively. While intraoperative applications of graph theory are under investigation, preoperative classification of hub nodes using centrality may serve as a useful tool to predict surgery outcomes, though a precise and validated workflow is required to utilize this in primary care. 

## Figures and Tables

**Figure 1 cancers-15-00556-f001:**
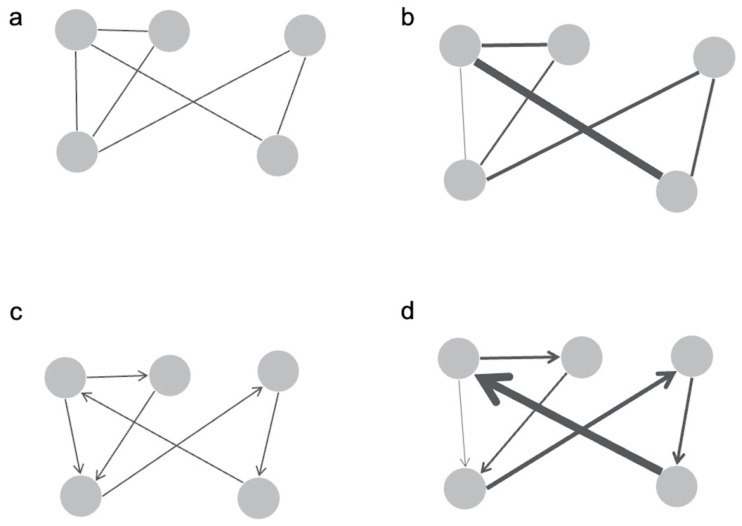
The circles represent network nodes, and the lines connecting the nodes are termed as edges. The edges of a network can be binary (**a**) or weighted (**b**). Thick bolded and highlighted lines represent weighted edges. Binary graphs may be useful when studying the topology of the network, while weighted graphs represent the diversity of connections between neural elements in the brain. Undirected straight-line connectors represent undirected edges, while the arrows represent directed edges. Edges may also be undirected (**a**,**b**) or directed (**c**,**d**) depending on the method used to construct the graph. In vivo human studies utilize undirected graphs. Note that these planar graphs are for visualization only, and real brain graphs do not necessarily have to be planar. Figure adapted from Farahani et al. 2019 [7].

**Figure 2 cancers-15-00556-f002:**
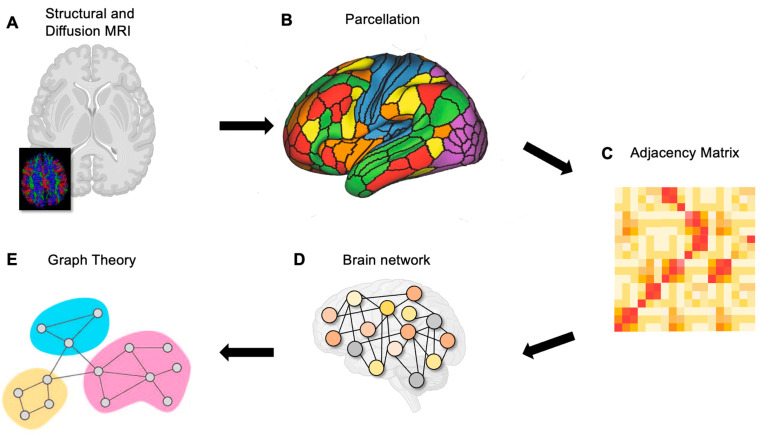
Schematic of brain network construction. (**A**) Neuroimaging data are used to estimate functional or structural connectivity using diffusion MRI, functional MRI, EEG, or MEG. (**B**) Parcellations are created based on T1 or T1-weighted structural MRI scans. A parcellation scheme is applied to define nodes. (**C**,**D**) The relationship between each pair of nodes is then represented as an adjacency matrix which can be used to construct a network of the brain. (**E**) Graph theory metrics can then be applied to the network to examine its topology. dMRI, diffusion magnetoencephalography; EEG, electroencephalography; MEG, magnetoencephalography; fMRI, functional magnetoencephalography. Figure adapted from Liao et al. (2017) [54].

**Figure 3 cancers-15-00556-f003:**
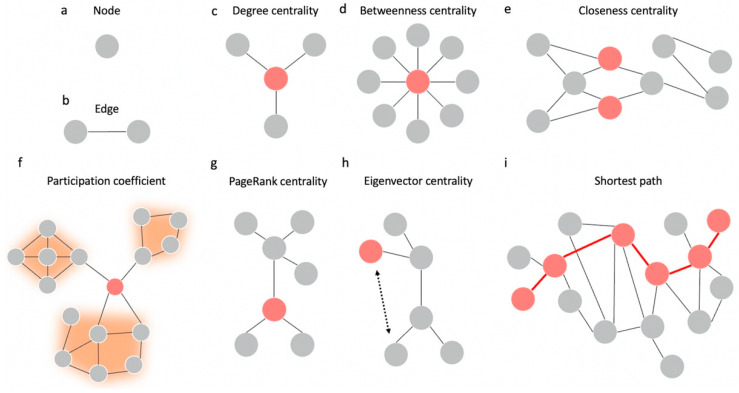
Visual representation of network centrality. In graphs of the brain, nodes (**a**) correspond to parts of the brain, while edges (**b**) signify either structural or functional connections. The nodes in red display the highest corresponding centrality. (**c**) The degree centrality is the number of direct neighbors a node has, such that the degree of the red node is 3. (**d**) The betweenness centrality measures the extent to which a node acts as a bridge between two other nodes. (**e**) Closeness centrality measures how fast a given node can access every other node in the graph. (**f**) Participation coefficient is a measure of the distribution of a node’s connections among the modules in a network, with each module represented in orange. (**g**) PageRank centrality scales the influence of a node’s neighbors by its degree. (**h**) Eigenvector centrality considers the quality of a node’s neighbors when quantifying its centrality. The red node has a higher eigenvector centrality than the grey node (pointed to by the dashed arrow), despite their degrees being the same. (**i**) Path length is a measure used to calculate the efficiency of information flow in a given network. Figure adapted from Farahani et al. (2019) [7].

**Figure 4 cancers-15-00556-f004:**
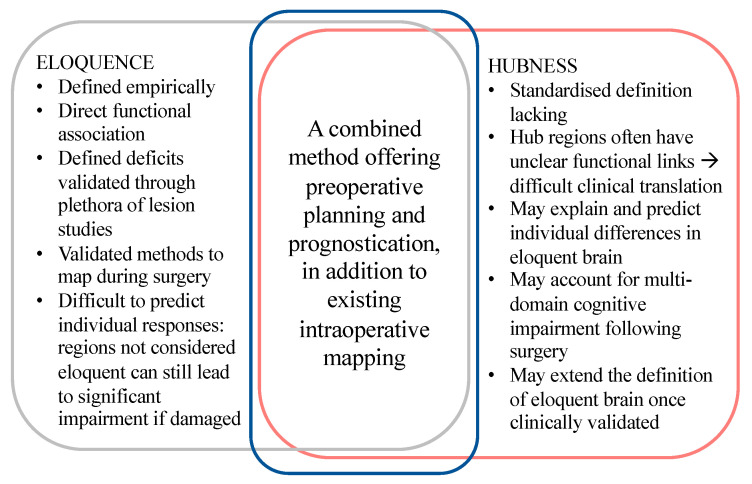
Comparing eloquence and hubness. Hubness may be used to extend the definition of eloquence once clinically validated. The intraoperative identification of eloquent brain can be combined with preoperative identification of hub nodes in each patient to further inform surgical decision-making.

**Table 1 cancers-15-00556-t001:** Glossary of network terms.

Term	Definition	Examples of Methods Used for Its Acquisition
Adjacency matrix	A summary of all connections between each pair of nodes.	--
Centrality	Measure of the importance of a node within the network.	Degree, betweenness, closeness, and PageRank
Connection	A relation or interaction between two nodes in the network. Connections may be binary or weighted and can be directed or undirected. They are referred to as edges in graphs.	Diffusion MRI (structural connectivity), functional MRI (functional connectivity)
Connectome	A map of all of the anatomical connections of the brain.	
Degree	The number of edges attached to a node.	Degree centrality
Edge	A term in graph theory to refer to a connection between two nodes.	Diffusion MRI (structural connectivity), functional MRI (functional connectivity)
Functional connectivity	The statistical correlation of co-activation of two nodes in the network.	fMRI, MEG, EEG
Graph	A mathematical representation of a network, comprising of nodes and edges.	--
Hub	A node with a central role in the network determined by its possession of links that greatly exceed the average, often defined through centrality.	Centrality
Module	A group of nodes within a graph which have many mutual connections, and few connections to nodes outside their module.	Number of links in the network; number of links between nodes in a specific module, summation of the degrees of the node within the module
Parcellation	An anatomical or functional division of the brain, which can be used as a node in graph theory.	Atlas-based schemes (Glasser, AAL, Gordon),
Participation coefficient	A measure of the distribution of a node’s edges across the modules within the graph.	Number of links to other nodes in a module; degree of the node
Path length	The number of edges which must be traversed to travel from one node to another node in the network. While this term technically refers to pathways in which the edges and nodes are traversed only once, it is commonly used in the literature to define any successive edges from one node to another. If there are multiple paths between two nodes, the path length may refer to the average length of all these paths.	The inverse of the average path length within the network, or the average distance between each pair of nodes, reflects the efficiency of information transferring in the entire network
Percolation	The method of deleting nodes within a network to model the effects of lesions on network topology.	Performed on an existing connectome to virtually emulate a lesion
Structural connectivity	The anatomical connections between the nodes of the network.	Diffusion MRI

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
