# Peer review of "Graph Theory Measures and Their Application to Neurosurgical Eloquence"

_cancers, 2023, doi:10.3390/cancers15020556_

Round 1
Reviewer 1 Report (Previous Reviewer 3)
Dear authors,
Thank your your revision.
But however, I am would like to suggest you reorganize the article to change your topic.
Regarding the title "Graph theory measures and their application to neurosurgical eloquence", readers would like to expect the applications based on each method you have mentioned, especially in neurosurgical domain. However, the application of each graph theory algorithm in neurosurgery and related studies have not been well elaborated.
So I'm sorry to say that this modification didn't live up to expectations.
Author Response
We thank the reviewer for their consideration.
Reviewer 2 Report (Previous Reviewer 2)
This review expounded the development and application of Graph Theory Measures in neurosurgery. The author described the difficulties of analysis and the prospects of clinical using in this edition. It could be accepted for publication. The number of subheadings should be checked.
Author Response
We thank the reviewer for their recommendation. We have corrected the subheading numbers, and amended the first mention of 6.2 to 6.1, as should have been the case.
Reviewer 3 Report (Previous Reviewer 1)
Improved manuscript, suited for publication
Author Response
We thank the reviewer for their consideration.
Reviewer 4 Report (New Reviewer)
I have the following comments on the paper under review:
In the second column of Table 1, either put a period ( . ) in all cells or remove the used ones.
In the third last cell of the second column of Table 1, revise the definition of "Path length"; the present definition has a flaw if there is more than one path between the considered nodes of a network.
From the schematic of brain network construction (Figure 2), it seems that the resulting graph is always a planar graph; if this is the case, I suggest adding a remark about it.
There are many topological indices, being studied in molecular graph theory, that may be used to predict different properties of the object corresponding to a graph; for example, see https://doi.org/10.1002/minf.201800008 and https://doi.org/10.1021/ja00856a001
What do you say about possible uses of such indices in neurosurgical eloquence?
Author Response
In the second column of Table 1, either put a period ( . ) in all cells or remove the used ones.
Author response: We thank the reviewer for their suggestion. We have now amended the second column to reflect this change.
In the third last cell of the second column of Table 1, revise the definition of "Path length"; the present definition has a flaw if there is more than one path between the considered nodes of a network.
Author response: We thank the reviewer for their suggestion. We have slightly amended the definition to reflect that there may indeed be more than one path between two nodes. The formal definition of a path is often differentiated from a walk and a trail in graph theory - this is however not commonly distinguished in most of the literature, especially in a clinical setting. We have therefore largely spared the reader from this discussion, and added that if there are multiple paths, the path length may then refer to the average length of all the paths between the two nodes.
From the schematic of brain network construction (Figure 2), it seems that the resulting graph is always a planar graph; if this is the case, I suggest adding a remark about it.
Author response: We thank the reviewer for their comment. The graphs in figure 2 were primarily for visualisation purposes, and their geometry in no way are supposed to reflect the complex geometry of brain graphs. Nonetheless, we have added a statement about their planar nature in the figure caption.
There are many topological indices, being studied in molecular graph theory, that may be used to predict different properties of the object corresponding to a graph; for example, see https://doi.org/10.1002/minf.201800008 and https://doi.org/10.1021/ja00856a001
What do you say about possible uses of such indices in neurosurgical eloquence?
Author response: We thank the reviewer for their insight. While we agree that this is an interesting concept worth exploring, we believe that the discussion of molecular graph theory and metrics such as the Zagreb index are outside the scope of this review and the readers' focus. We have nevertheless included some discussion about how it may be worth future studies to explore whether other metrics used in other fields may be applied to brain networks and provide clinically meaningful concepts (lines 285-288)
This manuscript is a resubmission of an earlier submission. The following is a list of the peer review reports and author responses from that submission.
Round 1
Reviewer 1 Report
This paper reviews advances in connectomic mapping to the application of individualized assessments of regions of eloquent brain function. This review provides a comprehensive overview of this topic. Although this type of work has been detailed elsewhere, this paper is unique it its application of these approaches to regions of eloquence for perioperative brain mapping. I suggest to add a few statements on challenges to implementation of these approaches in real-world settings and standardization of application of graph theory metrics in this context.
Reviewer 2 Report
This review summarized the current research surrounding novel graph theoretical concepts of hubness, centrality and eloquence and specifically its relevance to brain mapping in neurosurgery. they described the principle of graphing brain structure and functional connectivity, introduced the application of brain node hubness in eloquence identification. Some details should be noted:
1. which account for x percent of connections in the brain? Line 75
2. Line 162 is a repeated sentence.
3. The technical defects and application prospects of graph theory measures was mentioned insufficiently.
4. How does the graph theory measures instruct resection areas during surgery cooperating with intraoperative navigation.
5. Different scheme for brain parcellating would bring different functional connection network, how to select an appropriate method for surgery application?
Reviewer 3 Report
The article is titled "Graph theory measures and their application to neurosurgical eloquence. The application of graph theory in neurosurgery is a very attractive topic, and the focus should be on the application and significance of these techniques in neurosurgery.
1 “This review aims to outline and review current research surrounding novel graph theoretical concepts of hubness, centrality and eloquence and specifically its relevance to brain mapping in neurosurgery.” What did the author mean “specifically its relevance to brain mapping in neurosurgery”? Please rephrase here. (Line 33-35)
2 Table 1.
2.1 Parameters or graph properties and methods used for acquiring them needed to be separated and clarified.
2.2 Path length, it is a measure of the efficiency of information transferring on a network, but not efficiency of network.
2.3 Based on Graphic Theory, a hub is a node with a number of links that greatly exceeds the average.
2.4 Please re-check the Table 1, not limited to terms and their definitions mentioned above.
3 Figure 2 need to be revised:
3.1 no subfigure A, B, C, and D were marked
3.2 Parcellation should be based on the structural MRI scans, for instance, T1 or T1-weighted, rather than the dMRI.
4 Regarding connectomics, connectome, and brain network, please clarify and specify their definitions in the article.
5 “Figure 1. The circles represent network hubs, and the lines connecting the hubs are termed as edges”. Is it hubs or nodes? (Line 98)
6 “They lack spatial precision since their sensors aggregate activity over large populations of neurons” (Line 112-113). Sensors cannot aggregate signals and can only receive aggregated signals.
7 In part 2.2 “2.2. Graphing functional connectivity”, the summarization on connectivity analysis of EEG and MEG was way too simple, should be enriched.
8 “Methods of delineating regions of the brain to define nodes, or parcellation, alters the network measures derived from the resulting graphs” (Line 128-129). Please rephrase.
9 “Traditionally, parcellations have been based on anatomical landmarks and cytoarchitecture [34]”. In the reference 34, it is “However, this tool does not alleviate the need for more sophisticated labeling strategies based on anatomical or cytoarchitectonic probabilistic maps.” Please clarify the parcellation based on cytoarchitecture in MRI images.
10 In Line 121-123, “Since it is an indirect measure, it is unclear whether an increase in the BOLD signal is due to excitation or inhibition in an area and can be confounded by variations in vasculature [27] and ageing [28].” later in Line 141-152, this fMRI-based graph parcellation is however recommended by the author. Please elaborate what are the advantages and disadvantages of fMRI-related brain structure image parcellation? How to evaluate the application of low structural resolution fMRI in identifying functionally relevant structures? Please add answers to the article.
11 “The machine learning tool used by the Glasser scheme is not currently publicly available and therefore only the group average from the Glasser study can be used. Global parcellations may however overlook important individual differences and therefore, subject specific application of machine learning tools is required to increase the accuracy of parcellation.” Line 153-157. Please elaborate in the text the reasons why machine learning can improve the accuracy of segmentation.
12 Line 159-162. “Once a network is constructed, measurements can be made on the topology of the network (Figure 2). Within a network, nodes which have structural or functional signifi- cance are referred to as network hubs [40]. Once a network is constructed, measurements can be made on the topology of the network (Figure 2).” Repeated sentences, please revise.
13 “Hubs can be defined based on various measurements on the network, though centrality is the most common.” Please rephrase.
14 According to reference, Freeman LC (1978) Centrality in social networks conceptual clarification. Social Networks 1: 215-239 doi:doi.org/10.1016/0378- 8733(78)90021-7. The author, Freeman, refers to "Degree of centrality" in the text. Please explain the source of the quotation at line 192, and please clarify "Degree of centrality" and "Degree centrality".
15 From lines 205 to 217, it is about efficiency and is not part of the centrality analysis. Please delete or start a separate subsection.
16 In Line 217, reference 51 mainly focused on using use the participation coefficient as the sole measure of node importance instead of “degree-based hubs” from centrality analysis. It should be clarified in the article.
17 While the authors describe the various methods of graph theory in the article, the paragraphs of each method should be followed by an introduction of how the protocol was applied to neurosurgery and its significance to neurosurgery.